# Ensemble estimates of global wetland methane emissions over 2000-2020

Zhen Zhang[1], Benjamin Poulter[2], Joe R. Melton[3], William J. Riley[4], George H. Allen[5], David J. Beerling[6], Philippe Bousquet[7], Josep G Canadell[8], Etienne Fluet-Chouinard[9], Philippe Ciais[7], Nicola Gedney[10], Peter O. Hopcroft[11], Akihiko Ito[12], Robert B. Jackson[13], Atul K. Jain[14], Katherine Jensen[15], Fortunat Joos[16], Thomas Kleinen[17], Sara Knox[18,19], Tingting Li[20], Xin Li[1], Xiangyu Liu[21], Kyle McDonald[15], Gavin McNicol[22], Paul A. Miller[23], Jurek Müller[16], Prabir K. Patra[24,25], Changhui Peng[26], Shushi Peng[27], Zhangcai Qin[28], Ryan M. Riggs[29], Marielle Saunois[7], Qing Sun[16], Hanqin Tian[30], Xiaoming Xu[14], Yuanzhi Yao[31], Xi Yi[27], Wenxin Zhang[22], Qing Zhu[4], Qiuan Zhu[32], Qianlai Zhuang[21]

[1]National Tibetan Plateau Data Center (TPDC), State Key Laboratory of Tibetan Plateau Earth System, Environment and Resource (TPESER), Institute of Tibetan Plateau Research, Chinese Academy of Sciences, Beijing, 100101, China

[2]NASA Goddard Space Flight Center, Earth Sciences Division, Greenbelt, MD, USA

[3]Climate Research Division, Environment and Climate Change Canada, Victoria, BC, Canada

[4]Climate and Ecosystem Sciences Division, Lawrence Berkeley National Laboratory, Berkeley, California, USA

[5]Department of Geosciences, Virginia Polytechnic Institute and State University, Blacksburg, VA, USA

[6]School of Biosciences, University of Sheffield, U.K.

[7]Laboratoire des Sciences du Climat et de l'Environnement, CEA, CNRS, UVSQ, universit& Paris-Saclay, Gif sur Yvette, France

[8]Global Carbon Project, CSIRO Environment, ACT 2601, Australia

[9]Earth Systems Science Division, Pacific Northwest National Laboratory, Richland, WA 99352, USA

[10]Met Office Hadley Centre, Joint Centre for Hydrometeorological Research, Wallingford, U.K.

[11]School of Geography, Earth & Environmental Sciences, University of Birmingham, U.K.

[12]National Institute for Environmental Studies, Tsukuba, Japan

[13]Department of Earth System Science, Woods Institute for the Environment, and Precourt Institute for Energy, Stanford University, Stanford, CA 94305–2210, USA

[14]Department of Atmospheric Sciences, University of Illinois, Urbana, IL 61821, USA

[15]Department of Earth and Atmospheric Sciences, City College of New York, City University of New York, NY, USA

[16]Climate and Environmental Physics, Physics Institute and Oeschger Centre for Climate Change Research, University of Bern

[17]Max Planck Institute for Meteorology, Hamburg, Germany

[18]The University of British Columbia, Vancouver, BC, Canada

[19]McGill University, Montreal, QC, Canada

[20]LAPC, Institute of Atmospheric Physics, Chinese Academy of Sciences, Beijing, 100029, China

[21]Department of Earth, Atmospheric, Planetary Sciences, Purdue University, West Lafayette, IN, USA

[22]Department of Atmospheric Sciences, University of Illinois, Chicago, IL, USA

[23]Department of Physical Geography and Ecosystem Science, Lund University, Sölvegatan 12, 223 62, Lund, Sweden

[24]Japan Agency for Marine-Earth Science and Technology (JAMSTEC), Yokohama, Japan

[25]Research Institute for Humanity and Nature (RIHN), Kyoto, Japan

[26]Department of Biology Sciences, University of Quebec at Montreal, C.P. 8888, Succ. Centre-Ville, Montreal, QC H3C 3P8, Canada

[27]Sino-French Institute for Earth System Science, Laboratory for Earth Surface Processes, College of Urban and Environmental Sciences, Peking University, Beijing 100871, China

[28]School of Atmospheric Sciences, Sun-Yat-Sen University, and Southern Marine Science and Engineering Guangdong Laboratory (Zhuhai), Zhuhai 519000, China

[29]Department of Geography, Texas A&M University, College Station, TX, USA

[30]Center for Earth System Science and Global Sustainability, Schiller Institute for Integrated Science and Society, Department of Earth and Environmental Sciences, Boston College, Chestnut Hill, MA 02467, USA

[31]School of Geographic Sciences, East China Normal University, Shanghai, China

[32]College of Hydrology and Water Resources, Hohai University, Nanjing, 210098, China

*Correspondence to*: Zhen Zhang (yuisheng@email.com)

**Abstract.** Due to ongoing climate change, methane ($CH_4$) emissions from vegetated wetlands are projected to increase during the 21st century, challenging climate mitigation efforts aimed at limiting global warming. However, despite reports of rising emission trends, a comprehensive evaluation and attribution of recent changes remains limited. Here we assessed global wetland $CH_4$ emissions from 2000 to 2020 based on an ensemble of sixteen process-based wetland models. Our results estimated global average wetland $CH_4$ emissions at $158\pm24$ (mean $\pm 1\sigma$) Tg $CH_4$ yr$^{-1}$ over a total annual average wetland area

of $8.0\pm2.0$ Mkm$^2$ for the period 2010-2020, with an average increase of 6-7 Tg $CH_4$ yr$^{-1}$ in 2010-2019 compared to the average for 2000-2009. The increases in the four latitudinal bands of 90°S-30°S, 30°S- 30°N, 30°N-60°N, and 60°N-90°N were 0.1-0.2 Tg $CH_4$ yr$^{-1}$, 3.6-3.7 Tg $CH_4$ yr$^{-1}$, 1.8-2.4 Tg $CH_4$ yr$^{-1}$, and 0.6-0.8 Tg $CH_4$ yr$^{-1}$, respectively, over the two decades. The modeled $CH_4$ sensitivities to temperature show reasonable consistency with eddy covariance-based measurements from 34 sites. Rising temperature was the primary driver of the increase, while precipitation and rising atmospheric $CO_2$ concentrations

played secondary roles with high levels of uncertainty. These modeled results suggest climate change is driving increased wetland $CH_4$ emissions and that direct and sustained measurements are needed to monitor developments.

# 1 Introduction

Wetlands are the largest single source in the global methane ($CH_4$) budget, representing ~25-35% of the total combined natural and anthropogenic sources (Kirschke et al., 2013; Saunois et al., 2016, 2020), with an uncertainty range of 100-230 Tg $CH_4$ $yr^{-1}$ (Cao et al., 1996; Gedney et al., 2004; Bousquet et al., 2006; Petrescu et al., 2010; Spahni et al., 2011; Melton et al., 2013; Bridgham et al., 2013; Bloom et al., 2017; Poulter et al., 2017). Covering 8-10% of the global land surface (Zhang et al., 2021a), wetland area is sensitive to climate variations (Zhang et al., 2018; Zhu et al., 2017). Over the last deglaciation, wetlands played an important role in driving the rise of atmospheric $CH_4$ concentrations (Hopcroft et al., 2017; Nisbet et al., 2023; Kleinen et al., 2023). In recent decades, wetlands have experienced unprecedented and ongoing changes, including continuous thawing of permafrost (Natali et al., 2019; Treat et al., 2018), land-use change (Fluet-Chouinard et al., 2023), a lengthening of the growing season in the Arctic (Arndt et al., 2019), and expansion in tropical areas due to enhanced precipitation (Fleischmann, 2023). Recent evidence from in situ measurements (Rößger et al., 2022), data driven estimates (Yuan et al., 2024; Ying et al., 2024), and satellite observations (Feng et al., 2022) suggests that these ongoing changes could enhance wetland $CH_4$ emissions and thus affect the trajectory of atmospheric $CH_4$ concentration. Furthermore, atmospheric $\delta^{13}C$-$CH_4$ records also show a trend toward increased depletion since the late 2000s (Lan et al., 2021; Nisbet et al., 2019), indicating that isotopically light biogenic sources, such as wetlands (Basu et al., 2022; Feng et al., 2022), agricultural, and waste sources (Schaefer et al., 2016; Zhang, et al., 2021b) have become dominant contributors to the rise in atmospheric $CH_4$. Current estimates of wetland $CH_4$ emissions (hereafter denoted as $eCH_4$) in response to climate change are projected to increase by up to 15-30% by 2050 (Koffi et al., 2020; Zhang et al., 2017), accounting for 25-40% of the pledged reduction in anthropogenic emissions (Shindell et al., 2019). These trends and projections suggest that the emerging wetland-$CH_4$ climate feedback that influences atmospheric $CH_4$ concentration requires a better understanding of long-term changes in $eCH_4$.

Directly diagnosing the variations and trends of $eCH_4$ at large scales is challenging. Site-level measurements, such as those from chamber and eddy covariance techniques, are useful for identifying underlying mechanisms and monitoring $CH_4$ fluxes at the landscape scale but are difficult to upscale due to large uncertainties in extrapolation and the high spatial heterogeneity of wetland $CH_4$ fluxes (Chu et al., 2021; Kuhn et al., 2021). Interpreting $eCH_4$ using satellite observations and inversions of atmospheric concentration data is also subject to uncertainties in anthropogenic sources, other natural sources, atmospheric chemistry, and model errors associated with atmospheric transport (Gatti et al., 2021; Gloor et al., 2021; Palmer et al., 2022; Patra et al., 2011; Zhang et al., 2021c). Global wetland models, integrated within land biosphere models, can serve to bridge our understanding of wetland $CH_4$ processes and diagnosing wetland $CH_4$ dynamics at large scales (Melton et al., 2013; Wania et al., 2013). These models provide mechanistic explanations for the causes of changes in $eCH_4$ dynamics. Furthermore, recent advances in wetland models (Arora et al., 2018; Kaiser et al., 2017; Shu et al., 2020; Grant 2017; Chang et al. 2020) show significant potential for improving our understanding of $eCH_4$ through the incorporation of complex biogeochemical processes.

Current studies have reached various conclusions on the change in eCH$_4$ over the last decades. Studies based on single biogeochemical models (Zhang et al., 2018; Zhu et al., 2017) suggest a significant increase in eCH$_4$ from 2000-2006 to 2007-2017, while atmospheric inversions (Zhang et al., 2021c; Yin et al., 2021; Basu et al., 2022; Feng et al., 2022) suggested even higher rate increases, from 2 Tg CH$_4$ yr$^{-1}$ yr$^{-1}$ to 3 Tg CH$_4$ yr$^{-1}$ yr$^{-1}$ during the post-2010 period. Poulter et al., (2017) reported no significant change between the 2000-2006 and 2007-2012 periods based on an ensemble of wetland models, while Saunois

et al. (2020) show a slight increase (~2 Tg CH$_4$ yr$^{-1}$) in average for 2007-2017 compared to the 2000-2006 level using a large set of wetland CH$_4$ models. However, these models demonstrate considerable differences in estimated eCH$_4$, both spatially and temporally (Ma et al., 2021; Parker et al., 2020; Chang et al., 2023), primarily due to the sensitivity of their estimations to the wetland areal extent, the implemented biogeochemical structures, and parameterizations. The multi-model ensemble approach is applied to increase the skill, reliability, and consistency of model forecasts, potentially offsetting individual model

errors (Schaefer et al., 2012). However, a recent study (Chang et al. 2023) found that down selecting atmospheric inversion and wetland model CH$_4$ predictions based on a comparison to eddy covariance data did not reduce uncertainty in global eCH$_4$ estimates. Therefore, it has become necessary to thoroughly evaluate the performance of these models using the most recent generation of wetland models against an increasingly dense network of observations (Delwiche et al., 2021; Knox et al., 2019) from eddy covariance sites.


Here we conducted ensemble simulations of 16 wetland biogeochemical models following a common modeling protocol to provide monthly integrated global eCH$_4$ for the period of 2000-2020, as part of the Global Carbon Project's Methane Budget activity. The inundation dynamics of each model were simulated using a model-specific prognostic hydrological modeling approach as well as a set of diagnostic satellite-driven simulations. A set of factorial simulations were carried out to isolate the

effects of temperature, precipitation, and rising atmospheric CO$_2$ concentration. The modeled temperature sensitivity was evaluated against the global eddy covariance database, FLUXNET-CH$_4$ (Delwiche et al., 2021; Knox et al., 2019), and a data-driven global wetland CH$_4$ upscaling dataset UpCH$_4$ (McNicol et al., 2023) based on FLUXNET-CH$_4$. In addition, we examined the changes in eCH$_4$ for the year 2020, which was characterized as an extremely warm and wet year with the highest growth rate of atmospheric CH$_4$ observed over the study period.


## 2 Methods

### 2.1 Wetland model ensemble

Sixteen wetland models participated in the ensemble simulations (Table S1). Wetland CH$_4$ models can be generally described as functions describing the biogeochemical processes that control CH$_4$ production and oxidation through methanogenesis and

methanotrophy, and the biophysical processes that regulate CH$_4$ transport from the soil to the atmosphere (Table S1). Methanogenesis in the models is linked to different proxies (e.g., carbon substrate, heterotrophic respiration, net primary

production) with a wide range of model complexity - more sophisticated models include wetland Plant Functional Types (PFTs) and explicitly simulate the processes of $CH_4$ production, consumption, and transport, while the simplified models use generalized empirical equations to simulate net fluxes without explicitly calculating individual components of the $CH_4$ flux.


Wetlands were defined as naturally vegetated forested and non-forested ecosystems with saturated/inundated areas, excluding coastal wetlands, cultivated wetlands such as rice paddies, and open water systems such as rivers, lakes, ponds, and reservoirs. A prognostic wetland inundation scheme and a diagnostic wetland dataset Wetland Area and Dynamics for Methane Modeling (WAD2M v2; Zhang et al., 2021a) are applied to identify the wetland areal dynamics. The prognostic wetland areal dynamics

were independently determined by each model's hydrological modules, which use water table depth or soil moisture, combined with sub-grid topographic conditions to determine saturated areas within a land surface grid-cell (Zhang et al., 2016; Xi et al., 2022). Among the participating models, there was a large variation in complexity and in the level of comprehensiveness with which wetland extent were characterized. The modules for simulating inundation ranged from simplified TOPMODEL approaches to more sophisticated representations of water-table variation, with the estimated magnitude being influenced by

the hydrologic schemes utilized and the sensitivities to precipitation. The prognostic modeled wetland extent showed large variability in estimated magnitude but was consistent with satellite-based inundation products in predicting different phases of inundation (Xi et al., 2022; Zhang, et al., 2021a). The ensemble mean of the modeled wetland extent is close to 7.5 $Mkm^2$ as estimated by WAD2M but higher than the 4.6 $Mkm^2$ by the satellite-based product Global Surface Water Extent and Dynamics version 2 (GIEMS2; Prigent et al., 2020). The modeled temporal variations in wetland areas have high correlations with the

satellite-based products for the temperate region and high latitudes (Fig. S1), except for the tropics. The modeled temporal variations in wetland areas show high correlations with satellite-based products for temperate regions and high latitudes (Fig. S1), except in the tropics. The limited agreement in the tropics may be due to the influence of aerosols and clouds on satellite-based measurements, as well as the process-based model's performance limitations in representing wetland areas. The diagnostic runs are exclusively used for temperature dependence calculations due to a discontinuity issue in the WAD2Mv2

over a few tropical hotspots, which affect a subset of wetland models that are particularly sensitive to inundation in the hotspots.

## 2.2 Modeling protocol and simulation setups

The modeling protocol aimed to provide wetland $CH_4$ fluxes and quantify the associated uncertainties arising from model differences, meteorological forcing, and wetland extent dynamics. To quantify meteorological forcing uncertainty, we used two climate inputs, a ground-based monthly climate dataset from the Climatic Research Unit (CRU) (Harris et al., 2014), up

to 2020 and a harmonized daily dataset from the Global Soil Wetness Project-3 GSWP3-W5E5 through the year 2019, which is a multiple-source-based daily dataset (Cucchi et al., 2020; Dirmeyer et al., 2006) used in the Inter-Sectoral Impact Model Intercomparison Project 3a (ISIMIP3a). For models that require 6-hourly meteorological forcings, a temporal-interpolation dataset CRU-JRA was applied based on the Japanese Reanalysis Agency (JRA55), aligned with CRU. The atmospheric $CO_2$ concentration values for 1861-2020 were obtained from the CMIP6 experimental protocol (Meinshausen et al., 2017).

Ancillary data, such as soil texture and $CH_4$-related parameter sets used model-specific inputs. All the models were run in 'natural vegetation' mode without transient effects of land use and land cover change. Methane oxidation in wetland soils was implicitly included in the estimate but the upland oxidative sink was not included as it was not part of the net wetland emissions calculations. Models included the spin-up period to pre-industrial conditions assuming net ecosystem exchange equilibrium before 1860 by recycling fixed $CO_2$ concentrations (1860 level of 286.42 ppm) and meteorology (1901-1920).

**2.3 FLUXNET-CH$_4$ and machine learning-based upscaling product UpCH$_4$**

FLUXNET-$CH_4$ is the first global dataset of $CH_4$ eddy covariance measurements that includes $\sim$ 80 sites globally, including different wetland types from peatlands (e.g. bog, fen), mineral wetlands (e.g. marsh, swamp), and rice paddies. For this study, a subset of natural freshwater wetland sites was selected for the analysis. All the eddy covariance measurements used in this study were gap-filled daily total fluxes filled using an Artificial Neural Network (ANN) approach (Knox et al., 2019). In

addition, a data-driven gridded dataset UpCH$_4$ (McNicol et al., 2023) for 2001-2018, which is based on 119 site-years of $CH_4$ fluxes from the FLUXNET-CH4 dataset, was applied in the comparison. This dataset used a random forest model to upscale ground-based eddy covariance $CH_4$ flux data and then was forced with globally-gridded predictor data and two wetland extent products, to predict wetland $CH_4$ emissions. The predictors included data sources from climate, biometeorological, and soil properties.

**2.4 Time series decomposition and statistical analyses**

To attribute the time series of global wetland $CH_4$ emissions to what we consider the dominant drivers of change (i.e., temperature, precipitation, and $CO_2$ concentration), we applied a multiple regression approach (Piao et al., 2013) to estimate the parameters of global wetland $CH_4$ sensitivity to climate drivers using the following equation:

$$y = \beta CO_2 + \gamma Tmp + \delta Pre + c + \varepsilon \tag{1}$$

where $y$ is the global annual total wetland $CH_4$ emission of each model from the transient run, or from the observation-based upscaling dataset UpCH$_4$, and Tmp, Pre, and $CO_2$ are the mean annual temperature, total annual precipitation, and mean atmospheric $CO_2$ concentration for that year, respectively. $\gamma$, $\delta$, $\beta$, and c are regression coefficients and $\varepsilon$ is the residual error term. The regression coefficients were calculated using a maximum likelihood estimate. Changes in other meteorological forcings may also influence the estimation of $y$. These confounding drivers, such as solar radiation and wind speed, although

they are considered to have minor impacts on the variations of $eCH_4$, were implicitly accounted for in the regression coefficients.

**2.5 Model factorial experiment**

To further separate the contribution of different controls on the change in methane emissions ($\Delta eCH_4$) by climate variations and rising $CO_2$, we used a subset of four models that conducted factorial experimental simulations by holding each factor

constant during part of the transient runs. This subset of the wetland models (i.e., ELM-ECA, LPJ-wsl, SDGVM, and VISIT) performed a set of factorial simulations to specifically attribute the effect of temperature, precipitation, and rising $CO_2$ concentration on wetland $CH_4$ fluxes with the climatology of 2000-2006 for 2007-2020. The simulations were performed by running the model keeping one-factor constant at a time to estimate the contribution of each component to the total range of variations (Table S2). For these factorial simulations, we evaluated the annual amplitude of wetland $eCH_4$ as a relative

percentage change to minimize the impacts of different modeling implementation choices, such as different input variables among models. The effect of the total changes on the relative change in amplitude was represented by the difference between the transient (one factor is time-varying) and baseline (static at 2000-2006 levels) runs. For simplicity, the relative contribution of a single driver to $eCH_4$ variations was quantified as the transient run minus the specific control run. To calculate the contribution of each driver using the subset of the models, we calculated weighting factors per year across the models, with

lower bias resulting in higher weight relative to the full ensemble mean using an inverse function.

## 2.6 Temperature dependence calculation

To further evaluate the response of $eCH_4$ to rising temperatures, we calculated the modeled seasonal $eCH_4$ temperature dependence, referred to as the apparent $Q_{10}$ metric at the locations of 34 FLUXNET-$CH_4$ sites. This seasonal $Q_{10}$ differs from the intrinsic $Q_{10}$ prescribed in the parameterization of respiratory processes in each model. Here it represents the overall

response of $eCH_4$ along geographic temperature gradients. The apparent $Q_{10}$ is defined as $eCH_4$ sensitivity to temperature change. We calculated apparent $Q_{10}$ based on $CH_4$ emitting strength over a standard wetland area, which was calculated as the $CH_4$ fluxes divided by inundated area on a per-pixel basis to exclude the effect of inundation dynamics. To derive the temperature sensitivity of $eCH_4$ at the soil or ecosystem level, we applied the following equation:

$$R(i) = R_b(i)Q_{10}^{\frac{T(i) - T_{ref}}{\Gamma}} \qquad (2)$$

where R(i) is the net wetland flux at the location of site $i$, $R_b(i)$ is the basal net $CH_4$ flux at the reference temperature $T_{ref}$, and T(i) is ambient temperature. The parameters $Q_{10}$, $\Gamma = 10°C$, and $T_{ref} = 15°C$ are all time-independent constants. The $Q_{10}$ acting on specific time scales can be obtained from $eCH_4$ at corresponding specific time scales (i.e., seasonal total and annual total) by fitting an exponential regression with modeled $eCH_4$ and air temperature from CRU or GSWP3-W5E5. To quantify the uncertainty in observed apparent $Q_{10}$, we employed 1000 sets of resampled FLUXNET-$CH_4$ observations generated based on

a Gaussian distribution. The uncertainty range in measured seasonal mean $CH_4$ fluxes was determined by aggregating the uncertainty of daily total fluxes obtained through ANN gap filling.

# 3 Results and Discussion

## 3.1 Changes in eCH4 during the period of 2000-2020

The multi-model ensemble based on the prognostic inundation schemes shows that the average annual global eCH4 over the period 2000-2020 was $156\pm24$ Tg CH4 yr$^{-1}$ (mean$\pm1\sigma$). The average annual eCH4 increased from $153\pm23$ Tg CH4 yr$^{-1}$ during 2000-2009 to $158\pm24$ Tg CH4 yr$^{-1}$ during 2010-2020. 15 out of 22 model simulations show significant positive linear trends ($p < 0.01$) with an ensemble mean increase rate of $0.6\pm0.3$ Tg CH4 yr$^{-1}$ yr$^{-1}$ over 2000-2020 (Fig. 1a; Table 1; Fig. S2). Differences in total annual emissions between the two sets of simulations driven by two different climate datasets CRU and GSWP3-W5E5, agree well in the magnitude of the annual anomalies. Notable eCH4 variations to climate events were observed, such as the rise during the 2010 La Niña (+5.2 Tg CH4 yr$^{-1}$) and the decline during the 2015 El Niño (- 4.6 Tg CH4 yr$^{-1}$) after removing the positive linear trends. The multi-model ensemble wetland eCH4 response to climate events is consistent with those reported by earlier studies (Zhang et al., 2018; Zhu et al., 2017) using single wetland models, indicating a modulation of the phase of eCH4 anomaly ($\Delta$eCH4) by the El Niño-Southern Oscillation. The model ensemble demonstrates a consistent increase in interannual variability (IAV) in $\Delta$eCH4 from $3.6\pm1.6$ Tg CH4 yr$^{-1}$ during 2000-2009 to $4.7\pm1.5$ Tg CH4 yr$^{-1}$ during 2010-2020, suggesting a potential increase in eCH4 variability under climate change.

The models consistently show that 2020 is the strongest positive anomaly year during 2000-2020, with a net increase of 2 [-2, 7] Tg CH4 yr$^{-1}$ (mean [min, max]) in 2020 compared to 2019. This positive anomaly in 2020 (Table 1) is broadly consistent with a recent study (Peng et al., 2022) that reported $6\pm2.3$ Tg CH4 yr$^{-1}$ based on simulations of two bottom-up models with different climate datasets. The discrepancy in estimated magnitude between the Peng et al. (2022) and our results are partly due to the parameterizations of CH4 module that causes lower annual magnitude in this study ($\sim 162\pm23$ Tg CH4 yr$^{-1}$ in 2020) compared to the Peng et al. (2022) study ($177\pm31$ Tg CH4 yr$^{-1}$ in 2020). Additionally, the precipitation inputs in the climate forcing used in this study show a lower positive anomaly ($\sim$ of 20 mm yr$^{-1}$ in CRU over global wetland) in precipitation in 2020 compared to the reanalysis-based estimates ($\sim$ 40-117 mm yr$^{-1}$ over global wetland used in the study by Peng et al., (2022), which leads to lower estimates of wetland area and consequently lower emissions in this study. Moreover, our model ensemble does not indicate a strong increase (-0.2[-1.5-0.7] Tg CH4 yr$^{-1}$) in eCH4 in Africa in 2020. This contrasts with recent atmospheric inversions (Feng et al., 2023; Qu et al., 2023), which suggest a large increase of 11-17 Tg CH4 yr$^{-1}$ above 2019 levels in African CH4 emissions for 2020. The estimated increase from these inversions is equivalent to 55%-85% of total eCH4 in Africa during 2010-2019 in our study (Figure 2). These discrepancies highlight the need for further studies to investigate the differences between these two approaches, including uncertainty in climate inputs in process-based bottom-up models and partitioning difference sources in atmospheric inversions.

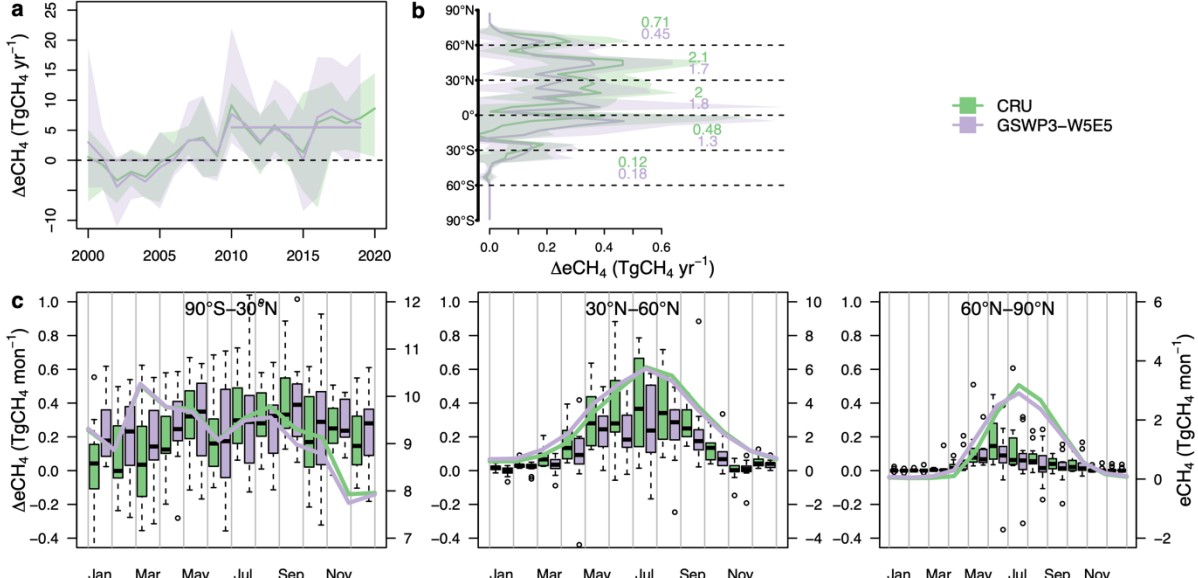

**Figure 1: Simulated model ensemble changes of global wetland CH₄ emissions for 2000-2020.** The change is expressed as the difference ($\Delta eCH_4$) relative to the mean of the 2000-2009 level from the two sets of simulations with prognostic wetland emission models grouped by different climate datasets, CRU and GSWP3-W5E5. a, Time series of annual total anomalies during 2000-2020, with the shaded area representing the range between minimum and maximum modeled emissions. The horizontal lines represent the ensemble means of 2000-2009 (152 Tg $CH_4$ $yr^{-1}$) and 2010-2019 (158 Tg $CH_4$ $yr^{-1}$), respectively. b, Latitudinal gradient of mean $\Delta eCH_4$, with the mean annual total $\Delta eCH_4$ for each of the 30° latitude bins from the two sets of simulations shown. c, Boxplots of mean seasonal $\Delta eCH_4$ for the three regions. The central mark and the bottom and top edges of the box indicate the median, and the 25th and 75th percentiles of the ensemble, respectively. The colored lines represent the average seasonal cycle of 2000-2009 from the simulations grouped by two climate datasets, CRU and GSWP3-W5E5.

There were widespread net increases in eCH₄ across all latitudinal bands during 2010-2020, compared to the average of 2000-2009, with the largest magnitudes occurring in the 90°S- 30°N bands (there are relatively few wetlands in the southern extra-tropics 90°S-30°S, contributing 0.1-0.2 Tg $CH_4$ $yr^{-1}$) and temperate regions (30-60°N) (Fig. 1b). The annual magnitude of eCH₄ increased by 3.7-3.8 Tg $CH_4$ $yr^{-1}$, 1.8-2.4 Tg $CH_4$ $yr^{-1}$, and 0.6-0.8 Tg $CH_4$ $yr^{-1}$ in the tropical, temperate, and Arctic wetlands, respectively. The tropics have experienced the largest increases in annual total emissions with an increase of 3% relative to 2000-2009 (Table 1). This finding is aligned with the results of several recent atmospheric inversions (Basu et al., 2022; Feng et al., 2022; Lan et al., 2021) using satellite observations and/or isotopic measurements that suggest a large increase in microbial emissions for post-2007 period in the tropics. While the increase in annual total emissions from temperate wetlands is lower than that from the tropics, they nevertheless show a larger relative increase of 5-8% compared to 2000-2009. Arctic wetlands also show an increased rate of 5-7% relative to the same period.

The increase in eCH₄ occurs in parallel with differing patterns of enhanced seasonal cycles between tropical and extratropical wetlands (30°N-90°N) (Fig. 1c). In temperate and Arctic wetlands, the majority of the increase in emissions (60-92%) occurred primarily during the growing season (May-October). Specifically, increases in Arctic wetlands occurred during the early

growing season (May-July), aligning with findings from a data-driven estimate (Yuan et al., 2024) and a long-term eddy covariance-based study (Rößger et al., 2022) that observed early growing season increases in eCH$_4$ due to continuous warming in a Siberian wetland. In contrast, the increase in emissions within the 90°S-30°N band exhibited relatively minor seasonal

variations throughout the year, with the May-October period accounting for a 24% greater increase in $\Delta$eCH$_4$ compared to the November-April period (Fig. S3).

**Table 1. Summary of wetland CH$_4$ emissions (Tg CH$_4$ yr$^{-1}$) over different time periods by latitudinal bands for the prognostic wetland simulations.** The ensemble mean with minimum and maximum (numbers within brackets) are listed, respectively.

| Time period | Forcing | 90°S-30°S | 30°S-30°N | 30°N-60°N | 60°N-90°N | Global |
|---|---|---|---|---|---|---|
| 2000-2009 | CRU | 3[1-5] | 107[63-141] | 31[16-60] | 11[4-29] | 152[119-187] |
| | GSWP3-W5E5 | 3[1-5] | 106[60-142] | 33[18-57] | 11[4-29] | 153[116-188] |
| 2010-2019 | CRU | 3[1-6] | 110[67-144] | 34[17-64] | 12[4-30] | 158[126-193] |
| | GSWP3-W5E5 | 3[1-6] | 110[64-146] | 35[18-60] | 12[4-29] | 158[118-203] |

**3.2 Spatial distribution of eCH$_4$**

A few key regions contribute significantly to global emissions (Fig. 2a,c). These regions are mainly floodplains located along major river basins such as the Amazon, Ganges, Mississippi, and Yangtze; tropical peatlands in the Congo and Southeastern Asia; and high-latitude peatlands in the Hudson Bay Lowland (HBL) and West Siberian Lowland (WSL). However, inter-model variabilities in eCH$_4$ reveal varying levels of spatial agreement between models, with the largest discrepancies coming

from South America and Africa. South America is one of the largest contributors to the global total eCH$_4$. Still, the net change in that region shows only a moderate increase, with diverging trends within the Amazon basin during the 2010s (Fig. 2b,d). The uncertain temporal trends are consistent with a long-term, large-scale atmospheric inversion based on airborne campaigns (Basso et al., 2021). South Asia and Africa are among the regions with the largest increases in the tropics, next to North America, but have high uncertainty with a lower level of agreement among the models (Fig. S4). The model ensemble shows

that Northwestern South Asia has a significant percentage increase in eCH$_4$ during 2010-2019 relative to its average levels from 2000-2009, suggesting a possible high sensitivity of eCH$_4$ to climate change in this region.

The comparison with previous estimates from bottom-up approaches and top-down atmospheric inversions (Table S3) suggests that the model ensemble mean generally captures well the spatial distribution of annual eCH$_4$, with a potential underestimation

for a few methane hotspots (Fig. S5). The model ensemble means for the Amazon basin, HBL, and WSL show good agreement with atmospheric inversions (Bergamaschi et al., 2013; Pickett-Heaps et al., 2011; Ringeval et al., 2014; Tunnicliffe et al., 2020; Wilson et al., 2016, 2021) and bottom-up modeling estimates (Bansal et al., 2023; Bloom et al., 2017; Bohn et al., 2015), with relatively low uncertainty. The model ensemble highlights WSL and HBL as $CH_4$ hotspots in the high latitudes, with good agreements of annual magnitudes with atmospheric inversions and in situ observations (Bohn et al., 2015; Glagolev et

al., 2011; Pickett-Heaps et al., 2011), while the models have lower estimates for Alaska compared to the inversions (Chang et al., 2014; Miller et al., 2016). However, for the two hotspots of the Pantanal and Sudd wetlands, the models tended to underestimate the annual $eCH_4$ compared to a few recent satellite-based estimates (Gerlein-Safdi et al., 2021; Gloor et al., 2021; Lunt et al., 2021; Pandey et al., 2021), with a large uncertainty range of up to two orders of magnitude across the model ensemble (Fig. S5). In addition to the regions where $eCH_4$ are being underestimated, recent studies (France et al., 2022; Shaw

et al., 2022) based on aircraft measurements suggest that the bottom-up models likely underestimate high $eCH_4$ fluxes in some little-studied wetlands, such as those in Zambia and Bolivia. The underestimations by process-based wetland models can be attributed to: 1) the challenge in accurately capturing the areal dynamics of wetlands under varying hydrological conditions, such as in flat terrains that receives lateral transport of water from upper streams (Li et al., 2024; Lunt et al., 2021; Gerlein-Safdi et al., 2021); 2) existing knowledge gaps in mapping wetlands in remote areas, which affect the parameterization of inundation modeling; 3) the limited representation of water table regulation (Chen et al., 2021) and wetland PFTs (Bastviken

et al., 2023) on $eCH_4$ in biogeochemical models.

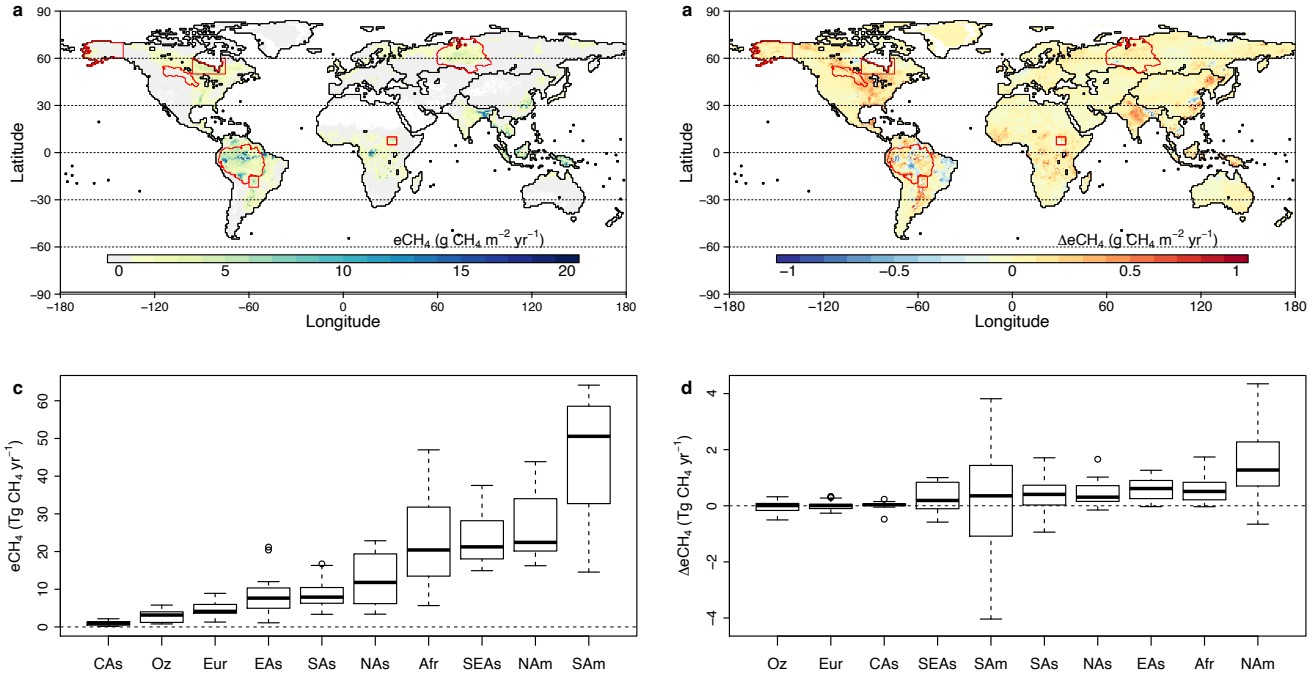

**Figure 2. Spatial distribution of $eCH_4$ and the average change between the 2010s and 2000s.** a. Map of mean $eCH_4$ (Unit: $gCH_4$ $m^{-2}$ $yr^{-1}$ per 0.5 deg grid cell) for 2000-2020. The regions defined in c, d and regional $CH_4$ hotspots in Table S3 are outlined in black and in red,

respectively. b. Map of change in mean annual wetland emissions ($\Delta eCH_4$) between the 2010s and 2000s. c. Boxplot of mean annual $eCH_4$ and d. $\Delta eCH_4$ by regions for 2000-2020 in ascending order for median estimates, Afr: Africa; CAs: Central Asia; EAs: East Asia; Eur: Europe; NAm: North America; NAs: North Asia; Oz: Oceania; SAm: South America; SAs: South Asia; SEAs: Southeast Asia.

**3.3 Attribution of wetland $CH_4$ changes**

To evaluate the relative contribution of different factors on global $eCH_4$, we calculated the sensitivity of $eCH_4$ to mean annual temperature (denoted as $\gamma$), annual total precipitation (denoted as $\delta$), and $CO_2$ concentration (denoted as $\beta$) using a multiple regression approach for each model run over the period of 2000-2020. The same approach was applied to the upscaled gridded machine learning dataset UpCH4, which uses eddy covariance measurements from FLUXNET-CH4 as training inputs. The model ensemble suggests that temperature is the primary driver of the increase in $eCH_4$ (Fig. 3a). The regression coefficients for $\gamma$ is 4.6 Tg $CH_4$ yr$^{-1}$ °C$^{-1}$, with a range of -0.4 and 9.0 Tg $CH_4$ yr$^{-1}$ °C$^{-1}$ between the 10th and 90th percentiles among all models. This mean temperature sensitivity is slightly higher than the $\gamma$ coefficient of 3.2-4.1 Tg $CH_4$ yr$^{-1}$ °C$^{-1}$ estimated for UpCH4. In contrast, precipitation contributed little to the increase from the prognostic simulations, with a coefficient $\delta$ of 0 to 0.3 Tg $CH_4$ yr$^{-1}$ mm$^{-1}$. The coefficient $\delta$ was lower at -0.05-0 Tg $CH_4$ yr$^{-1}$ mm$^{-1}$ for UpCH4, as precipitation was not chosen as a model training predictor through its feature selection, based on site-level eddy covariance measurements (McNicol et al., 2023). However, precipitation is a more dominant factor at large scales, especially for tropical floodplains, which contribute the largest proportion of emissions but are poorly represented by eddy covariance measurements. The model ensemble estimated $\beta$ remains small, ranging from 0 to 0.3 Tg $CH_4$ yr$^{-1}$ ppm$^{-1}$, while UpCH4 suggests a $\beta$ at -0.01 Tg $CH_4$ yr$^{-1}$ ppm$^{-1}$. However, other confounding drivers might influence $eCH_4$ as well, such as solar radiation, wind speed, and nitrogen deposition. Thus, the inferred sensitivities are implicitly accounted for in the regression coefficients despite their relatively small impacts compared to the major drivers.

Generally, the factorial simulations of the four-model subset indicated a consistently positive contribution (three out of four) from rising temperature to $\Delta eCH_4$, with a large variability (s.d.=4.3 Tg $CH_4$ yr$^{-1}$) of contributions from precipitation (Fig. S6). The strength of the $CO_2$ fertilization effect varied among models and was moderate but positive in all models. Two models (ELM-ECA and SDGVM) were among the models with higher sensitivity to climate variations while LPJ-wsl and VISIT were close to the full ensemble mean. ELM-ECA produced a negative temperature effect on $eCH_4$, likely due to its modeled nutrient constraints and higher temperature sensitivity for methanotrophic compared to methanogenic processes. Considering the deviation of each model from the full ensemble mean, the weighted mean (Fig. S7) contributions for temperature, precipitation, and $CO_2$ concentration from the subset models were 3.2, 1.8, and 1.4 Tg $CH_4$ yr$^{-1}$, respectively. The results from the subset of the models consistently demonstrate that temperature is the primary factor influencing $eCH_4$.

Overall, the interannual variations of modeled $eCH_4$ were primarily associated with rising temperature, altered precipitation patterns, and rising atmospheric $CO_2$ concentrations that stimulated ecosystem productivity through the $CO_2$ fertilization effect (Yvon-Durocher et al., 2014). We note that a recent study found strong hysteresis in the seasonal temperature dependence of observed $eCH_4$ using the FLUXNET-$CH_4$ dataset (Chang et al. 2021). Those hysteretic features likely result in uncertainty in annual temperature sensitivity estimates but would not bias the conclusion of temperature as a dominant controller of $eCH_4$ at the decadal time scale. The links between rising temperature and enhanced net $CH_4$ fluxes are evident (as described below), as the annual global average temperature over wetland areas has significantly ($p < 0.01$) increased by 0.5-0.7 °C from 2000-2020 (Fig. 3b). The modeled interannual variations of wetland extent dynamics reproduced the response to strong climate events (e.g., positive anomaly during the La Niña phase in 2010/2011 (Boening et al., 2012) and 2020). Both climate-forcing datasets suggest no significant trend in the anomaly of annual mean wetland area globally over the same period based on the prognostic hydrological simulations (Fig. 3b). Similarly, no significant regional trends in wetland area were found for most of the sub-regions, with the exception of South America, which shows a decrease, and East Asia, which shows a slight increase (Fig. S8). Considering that the extent of modeled wetland areas is primarily driven by precipitation, we do not detect a substantial contribution of changes in wetland extent to the long-term increase in $eCH_4$ over 2000-2020 based on the climate datasets. However, considerable differences in annual and seasonal precipitation estimates between the climate datasets used in this study and those derived from reanalysis or satellite-based products (Zhang et al., 2023a) result in large uncertainties in the estimated trends in wetland extent.

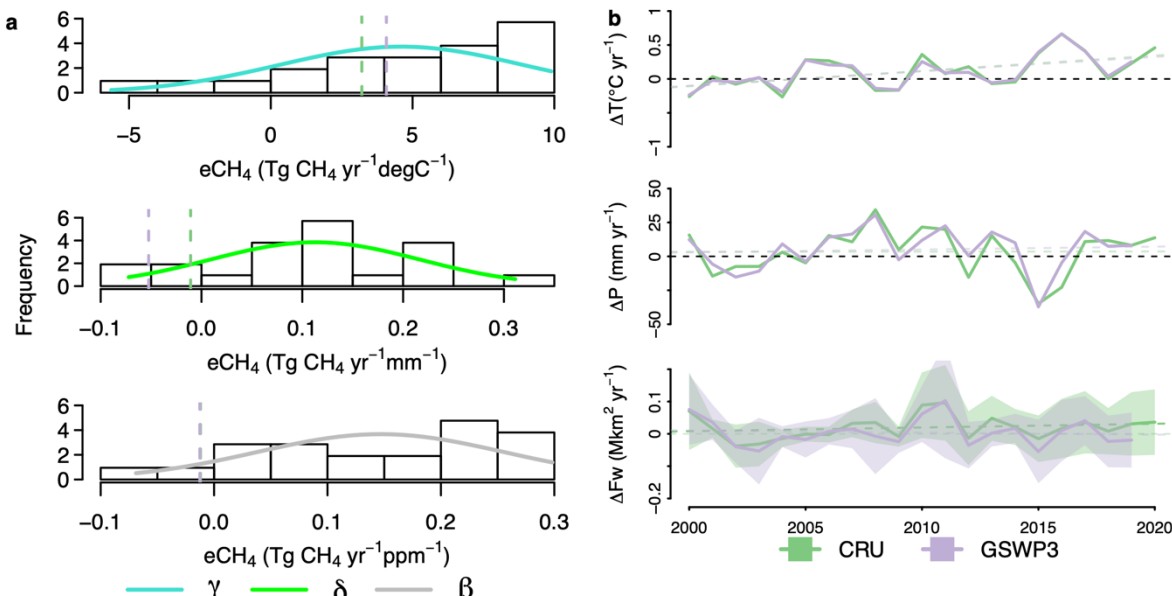

365

**Figure 3. Attributions of ΔeCH₄ during 2000-2020.** a. Histogram showing the sensitivity coefficients derived from a multiple regression approach (See Methods) for temperature ($\gamma$), precipitation ($\delta$), and atmospheric $CO_2$ concentration ($\beta$). The curves represent probability distributions of sensitivity coefficients across the models, assuming a Gaussian distribution. Vertical lines represent estimates from the machine learning-based dataset UpCH₄, with different colors corresponding to different climate datasets. b. Time series of anomalies for annual mean temperature ($\Delta T$), annual total precipitation ($\Delta P$), and annual mean wetland extent ($\Delta Fw$) for 2000-2020 for CRU and 2000-2019 for GSWP3. The shaded area in $\Delta Fw$ represents the minimum and maximum range from the prognostic model simulations. Dashed lines are linear fitted trends for corresponding variables.

## 3.4 Temperature sensitivity of wetland CH₄ models

The modeled CH₄ emissions show an exponential relationship between eCH₄ and air temperature, with higher temperatures corresponding to higher mean eCH₄ during the peak growing season (JJA, June-July-August) in the Northern Hemisphere (Fig. 4a). The model ensemble mean of eCH₄ response to temperature shows good agreement within the range of the spread when compared to the site-level measurements from FLUXNET-CH₄ and the gridded product UpCH₄. The model ensemble mean has a higher CH₄ emitting strength (i.e., CH₄ emission per standard wetland area) for the high latitudes, leading to lower apparent $Q_{10}$. This implies that the model ensemble estimated temperature sensitivity for the high latitudes could be potentially overestimated during the JJA season. The apparent $Q_{10}$ values for individual models show a large spread (Fig. S9), with eleven out of the sixteen models having statistically significant ($p < 0.01$) exponential relationships. The good agreement between the ensemble mean and observations suggest that the ensemble approach provides a better constraint compared to single models alone. Furthermore, it is important to acknowledge that the sparse spatial coverage of FLUXNET-CH₄ over low latitudes, especially for underrepresented areas such as Africa, Southeast Asia, and South America, limits our ability to evaluate temperature dependencies over high-temperature regions (Fig. S10).

The modeled apparent $Q_{10}$ exhibits an average temperature dependence similar to that of ecosystem respiration, as reported by previous studies (Bloom et al., 2017; Mahecha et al., 2010; Yvon-Durocher et al., 2014), indicating that the underlying factors controlling the response of eCH₄ and ecosystem respiration to temperature covary. The modeled temperature dependences are more constrained with less spread for JJA and SON (September-October-November) than DJF (December-January-February) and MAM (March-April-May) when most site-level measurements have limited availability. The seasonal variations of modeled apparent $Q_{10}$ differ from site-level observations or UpCH₄, reflecting discrepancies in the involved processes between eddy covariance and land surface models. Given that underrepresented processes such as substrate supply tend to have higher sensitivity of ecosystem metabolic processes to temperature, it is likely that the models do not entirely capture the fine-scale processes that affect the overall temperature response (Chang et al. 2021). In addition, the absence or underrepresentation of certain biophysical processes could lead to lower modeled apparent $Q_{10}$. For instance, the ensemble mean of modeled apparent

Q$_{10}$ for SON seasons is underestimated, likely linked to the limited representation of processes during the freeze/thaw cycle (e.g., zero-curtain period), as suggested by previous observational studies (Mastepanov et al., 2008; Zona et al., 2016).

400

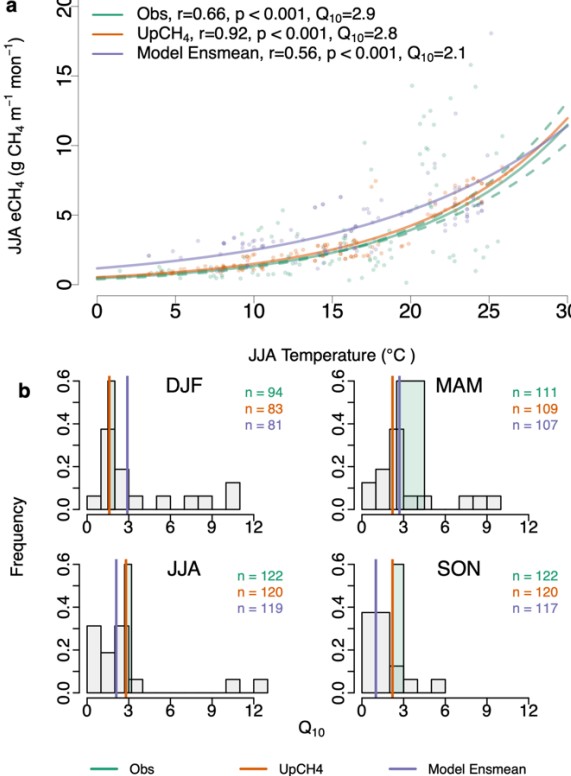

**Figure 4. Temperature sensitivity of simulated seasonal eCH$_4$ across locations of FLUXNET-CH4 sites.** a. Model ensemble mean ('Model Ensmean') of simulated eCH$_4$ against seasonal mean temperature for the JJA season along the temperature gradient at the locations of FLUXNET-CH$_4$ sites in comparison to the estimates from eddy covariance measurements ('Obs'; Fig. S10; Table S4) and UpCH$_4$. Each
405    dot represents the value at one site for an individual year when observations are available. The unit of the simulated CH$_4$ emissions is g CH$_4$ m$^{-1}$ month$^{-1}$ per standard wetland area to exclude the effect of inundation on eCH$_4$. The exponential fitted curves are shown. b. Histogram of the seasonal Q$_{10}$ for the 16 individual models for the months DJF, MAM, JJA, and SON. Sample sizes are shown in the plot. The Q$_{10}$ values derived from FLUXNET-CH$_4$, UpCH$_4$, and the model ensemble mean are vertical solid lines, with a width of the bar for 'Obs' indicating the uncertainty range of Q$_{10}$ based on measurement uncertainty.

410

**4 Conclusions**

Our results estimated global average wetland CH$_4$ emissions at 158±24 (mean ± 1$\sigma$) Tg CH$_4$ yr$^{-1}$ for the period 2010-2020, with an average decadal increase of 6-7 Tg CH$_4$ yr$^{-1}$ compared to the decade of 2000-2009. The increases in the four latitudinal

bands of 90°S-30°S, 30°S- 30°N, 30°N-60°N, and 60°N-90°N were 0.1-0.2 Tg $CH_4$ $yr^{-1}$, 3.6-3.7 Tg $CH_4$ $yr^{-1}$, 1.8-2.4 Tg $CH_4$ $yr^{-1}$, and 0.6-0.8 Tg $CH_4$ $yr^{-1}$, respectively, during the two decades. Our analysis reveals how global wetlands respond to variations in the primary climatic controls of temperature, precipitation, and rising $CO_2$ concentrations. The model average shows good agreement with eddy covariance measurements on temperature dependence, confirming the primary role of temperature in the rising trajectory of $eCH_4$ at decadal timescales. Furthermore, the modeled ensembles of prognostic wetland extents offer a complementary approach to satellite-based estimates (Prigent et al., 2020; Zhang, et al., 2021a) and enable further investigation into the uncertainties in wetland area estimation. These differences can motivate improvements to inundation schemes through an improved water table position (Chen et al., 2021) and lateral flow representation. Note that a large portion of tropical wetlands comprises inundated floodplains connecting rivers, where the leaching of methane production from wetlands to river networks is not accounted for in the wetland models. The prognostic models estimate an annual mean maximum wetland area of 8.0±2.0 $Mkm^2$, with a seasonal cycle (annual maximum minus annual minimum) of 4.7±2.0 $Mkm^2$. Resolving the large uncertainty in wetland areas and seasonal variation remains a high priority to refine bottom-up estimates of $eCH_4$. Lastly, our results highlight the important but highly uncertain $CO_2$ fertilization effect on $eCH_4$. The mean sensitivity coefficient $\beta$ and results from the factorial experiment suggest a net increase of $eCH_4$ of 0.1%-2.3% relative to the annual total under an average ~20 ppm increase in atmospheric $CO_2$ concentration. In comparison, a synthesis study based on field experiments (van Groenigen et al., 2011) shows a narrower range of 0.3%-0.6% average increase for every 20 ppm increase, assuming a linear fertilization effect between $CO_2$ concentration and $eCH_4$.

Our results show that an ensemble of process-based wetland methane models provides quantification for uncertainty in $eCH_4$, as well as better constraints than a single model on the predicted trend and magnitude of $eCH_4$. However, nominally distinct models might have similar biases because of similarities in the way they represent a subset of processes (see Table S1 for the model summary). Future evaluation of modeled processes, such as oxidation, production, and transport pathways, along with model error across different time scales using statistical tools could help identify similarities in model behaviors to guide model development (Zhang, 2023b). Furthermore, the $eCH_4$ estimates are subject to forcing uncertainty, given that the two climate datasets applied in the simulation protocol do not cover the full magnitude and variability of climatic variables. Specifically, precipitation has a significant impact on wetland extent and anaerobic soil conditions but has large uncertainty in spatiotemporal patterns (Sun et al., 2018). Thus, we recommend future ensemble simulations consider the uncertainty in climate variables among different datasets. In addition, the sensitivity parameters derived from the multiple regression are not independent of climate datasets. Thus, they are affected by the choice of meteorological drivers. Overall, quantitatively accounting for model performance and dependence and thoroughly evaluating the effectiveness (Chang et al., 2023) could improve the wetland model ensemble estimation in future studies.

**Code and data availability**

The code for the wetland models is available upon request from the respective model groups. The wetland ensemble results is publicly available at the Zenodo Repository 10.5281/zenodo.11309188. The wetland estimates from individual models are available upon request from respective model groups. The FLUXNET-CH$_4$ dataset is publicly available at the link:
https://fluxnet.org/data/fluxnet-ch4-community-product/. The UpCH$_4$ dataset can be found at the link in McNicol et al., (2023).

## Author contribution

BP and ZZ designed the simulation experiment with contributions from JM and WR. ZZ conducted data collection and data
analysis. JM, WR, GB, PC, NG, PH, AI, AJ, FJ, TK, TL, XL, PM, JM, CP, SP, ZQ, QS, HT, XX, YY, XY, WZ, QZ, QZ, QZ, and ZZ performed the simulations. ZZ prepared the manuscript with contributions from all co-authors.

## Competing interests

At least one of the (co-)authors is a member of the editorial board of Biogeosciences.

## Acknowledgments

This paper is the result of a collaborative international effort under the umbrella of the Global Carbon Project, a project of Future Earth, and a research partner of the World Climate Research Programme. Z.Z acknowledge support from National Natural Science Foundation of China Basic Science Center for Tibetan Plateau Earth System project. X. Y and S. Peng were
funded by NSFC (41830643, 41722101). Thomas Kleinen acknowledges support from the German Federal Ministry of Education and Research (BMBF), Grant No. 01LP1921A. J.R.M. thanks Jade Skye for her assistance in running and processing the CLASSIC simulations. A. Ito was partly supported by MEXT Arcs-II. L. Liu and Q. Zhuang are supported by NASA project (NNX17AK20G). Q. Zhu and C. Peng are supported by the Second Tibetan Plateau Scientific Expedition (2019QZKK0304). J. Müller and F. Joos were supported by the Swiss National Science Foundation (#200020_200511). Q.
Zhu and W. Riley were supported by the Reducing Uncertainties in Biogeochemical Interactions through Synthesis and Computation (RUBISCO) Scientific Focus Area and Energy Exascale Earth System Modeling Project, which are sponsored by the Earth and Environmental Systems Modeling (EESM) Program under the Office of Biological and Environmental Research of the U.S. Department of Energy Office of Science. Y. Yao and H. Tian are funded in part by NSF program (award numbers: # 1903722) NASA CMS Program (award numbers: NX14AO73G). T. Li was supported by the National Key
Scientific and Technological Infrastructure project "Earth System Science Numerical Simulator Facility" (EarthLab) and the Open Research Program of the International Research Center of Big Data for Sustainable Development Goals (Grant No. CBAS2023ORP02). P. Hopcroft was supported by a Birmingham Fellowship and the University of Birmingham's BlueBEAR HPC service. W.Z. acknowledges the support from the LUNARC computation project LU 2021/2-114 and the Swedish Research Council (Vetenskapsrådet) starting grant 2020-05338. W.Z. and P.A.M. acknowledge this study as a contribution to

the strategic research areas Modeling the Regional and Global Earth System (MERGE) and Biodiversity and Ecosystem Services in a Changing Climate (BECC) at Lund University. RB Jackson acknowledges support from the United Nations Environment Programme (UNEP) to Stanford University DTIE21-EN3143. N.G. was supported by the Newton Fund through the Met Office Climate Science for Service Partnership Brazil (CSSP Brazil). A. Jain and X. Xu were supported by the US National Science Foundation (NSF- 831361857) and would like to acknowledge the high-performance computing support

from Cheyenne (doi:10.5065/D6RX-99HX) provided by NCAR's Computational and Information Systems Laboratory, sponsored by the National Science Foundation. P.C. acknowledges support from the space Agency Climate Change Initiative (ESA CCI) RECCAP2 project (grant no. ESRIN/4000123002/18/I-NB). JG Canadell acknowledges the support of the Australian National Environmental Climate Science Program - Climate Systems hub. G. McNicol acknowledges support from the NASA CMS program (award number: NNH20ZDA001N).

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
