# Peer review of "Ensemble estimates of global wetland methane emissions over 2000-2020"

_EGUsphere, 2024_

## Author Comment (AC1)

RC1

Review Comments for egusphere-2024-1584
Title: Ensemble estimates of global wetland methane emissions over 2000-2020

This is a comprehensive analysis of global inland wetland CH4 emission study that has taken a lot of efforts from many field observations and models. The final result of 158 Tg annual CH4 emission is an important number that fills the gap of current carbon cycle science. I would suggest acceptance after a minor revision.

Response: Thank you for your constructive comments. We appreciate your recognition of the significance of our findings.

Abstract:
It would be good to add the global total wetland area in Abstract.

Response: We have added global total wetland area in the Abstract as follow: *"Our results estimated global average wetland CH$_4$ emissions at 158±24 (mean ± 1$\sigma$) Tg CH$_4$ yr$^{-1}$ over a total annual average wetland area of 8.0±2.0 Mkm$^2$ for the period 2010-2020".*

Introduction:
Although a lot of references was mentioned, I didn't see a number or a range of global annual CH4 emission provided by previous studies. If such numbers exist, please try to add the information.

Response: We have modified the introduction to include a range of global annual CH4 emissions as follow:

"Wetlands are the largest single source in the global methane (CH$_4$) budget, representing ~25-35% of the total combined natural and anthropogenic sources (Kirschke et al., 2013; Saunois et al., 2016, 2020), with an uncertainty range of 100-230 Tg yr-1 (Cao et al., 1996; Gedney et al., 2004; Bousquet et al., 2006; Petrescu et al., 2010; Wania et al., 2010; Spahni et al., 2011; Melton et al., 2013; Bridgham et al., 2013; Bloom et al., 2017; Poulter et al., 2017)."

Methods:
Line 140-143: wetland extent
It would be good to add global wetland area here, or the range of wetland area from those models, or the area from GIESM2.

Response: We have added the wetland area numbers in the sentence as follow:

*"The ensemble mean of the modeled wetland extent is close to 7.5 Mkm2 as estimated by WAD2M but higher than the 4.6 Mkm2 by the satellite-based product Global Surface Water Extent and Dynamics version 2 (GIEMS2; Prigent et al., 2020). The modeled temporal variations in wetland areas have high correlations with the satellite-based products for the temperate region and high latitudes (Fig. S1)."*

Line 156: Ancillary data.
It would be good to list a few more data items beyond climate data and soil wetness data, e.g., some soil carbon data and vegetation type. I understand those data maybe quite different among the 16 models. Just a suggestion, not a must. Also, maybe list a few CH4-related parameters that most models have in common?

Response: Thank you for the suggestions. Given the level of different complexity and structure in the CH4 modules, it could be potentially misleading if we directly compare the values of parameters among models. Thus we are unable to provide specific values of parameters. Instead, we have listed descriptions about how model implement transport pathways, $CH_4$ production proxy, and temperature response functions in the Supplement Table S1.

Results:
Figure 1. Nice. But I only see delta CH4 values. Maybe the mean value of the 2000-2009 level should be added in the figure caption? What if you plot the absolute CH4 values on panel 'a'? Should they be the same curves but vary around the mean value?

Response: Thank you for your suggestions. We have added the ensemble mean value for 2000-2009 to the caption of Figure 1(a) as follows: "The horizontal lines represent the ensemble means of 2000-2009 (152 Tg CH4 yr$^{-1}$) and 2010-2019 (158 Tg CH4 yr$^{-1}$), respectively." We have included the absolute CH4 values in the Supplement, as showing the changes with absolute values in the main figure is not as visually clear.

Figure 2 a. Visually, I guess panel 'a' can be improved with a more contrasting color scheme. It will be interesting to see the spatial variation on those low value areas. Currently they are all yellow.

Response: We have modified the color scheme to make the plot more color-contrasted.

Line 313: Is it multiple liner regression? If so, add the word 'liner'.

Response: Added.

Figure 3 a. Panel 'a' comparing histogram (frequency) against degree/mm/ppm is unfamiliar to me. No critics here, just make sure you can explain well "The model ensemble suggests that temperature is the primary driver of the increase in eCH4 (Fig. 3a)." What if you use precipitation unit in cm, will that change anything? (I am not quite good at statistics.)

Response: Thank you for the suggestion. Changing the unit for precipitation to centimeters, or for $CO_2$ concentration, would not alter the pattern of the histogram, as the multiple regression is independent of the units used for precipitation.

Another way to explain why temperature is the primary driver of CH4 increase may be in panel 'b'. I see panel 'b' has regression trend lines, which may show delta T-CH4 relationship is significant because the slope seems bigger. Maybe precip and CO2 impacts are statistically **insignificant**?

Response: Thanks for the suggestion. Yes we agree with the suggestion. The panel 'b' shows that temperature has a significant increasing trend while the trends for change in precipitation and wetland area globally is not statistically significant. We have descriptions in the text as follow: "The links between rising temperature and enhanced net CH4 fluxes are evident (as described below), as the annual global average temperature over wetland areas has significantly ($p < 0.01$) increased by 0.5-0.7 °C from 2000-2020 (Fig. 3b). The modeled interannual variations of wetland extent dynamics reproduced the response to strong climate events (e.g., positive anomaly during the La Niña phase in 2010/2011 (Boening et al., 2012) and 2020). Both climate-forcing datasets suggest no significant trend in the anomaly of annual mean wetland area globally over the same period based on the prognostic hydrological simulations (Fig. 3b)."

Line 366: There are two section '3.2'. The former is on Line 310.

Response: Fixed.

Conclusions:
Line 416-417: "Resolving the large uncertainty in wetland areas and seasonal variation remains a high priority to refine bottom-up estimates of eCH4."
Is it possible to report the wetland areas and/or seasonal variation from the 16 models?

Response: We have added one sentence in the conclusion as follow: *"The prognostic models estimate an annual mean maximum wetland area of 8.0±2.0 Mkm2, with a seasonal cycle (annual maximum minus annual minimum) of 4.7±2.0 Mkm2."*

---

## Author Comment (AC3)

RC2

In "Ensemble estimates of global wetland methane emissions over 2000-2020," Zhang and co-authors simulate global methane wetland emissions using 16 process-based wetland models with varying levels of complexity that are participating in the Global Carbon Project. Authors simulate wetland methane emissions for 2000-2020 and simulate the decadal changes in emissions and their large-scale drivers. The modeling ensemble shows an increase in 2010-2020 vs 2000-2010 emissions, and that temperature is the primary driver followed by precipitation and atmospheric CO2 concentration. Authors show that these changes and the drivers are generally support by inversions and observational evidence.

Overall, I think this is a well written study and useful study. In my opinion, it should be accepted after addressing a few comments and questions.

Response: Thank you for your constructive feedback. We appreciate your acknowledgment of the importance of our findings.

Overall comments

The multiple linear regression lacks detail in how the predictors were selected, so it is unclear how robust those conclusions are. Authors choose global mean temperature, global total precipitation, and mean atmospheric CO2 concentration as the predictors, and then state that modeled eCH4 was "primarily associated" with those variables (line 340), but were those the only variables tested? In that case, did the exercise reveal anything new? Authors say that "other confounding drivers might influence eCH4 as well, such as solar radiation, wind speed, and nitrogen deposition" (line 325), but don't explore these as predictors. Could authors provide more justification for their choice of the three main predictors? Did authors test model performance after leaving any of these predictors out, or adding any of the additional predictors they mentioned?

Response: Thank you for your insightful feedback. We appreciate your questions regarding the selection of predictors in our multiple regression analysis. The choice of global mean temperature, total precipitation, and atmospheric $CO_2$ concentration as predictors was based on findings from previous studies (Piao et al., 2013; Zhao et al., 2016), which identified these variables as dominant drivers of carbon fluxes in process-based models. We acknowledge that other meteorological factors, such as solar radiation, wind speed, and nitrogen deposition, may also influence eCH4. However, it is important to note that only a few models currently implement wind speed as inputs

and the nitrogen cycle, making it practical to focus on the three primary factors. While these additional factors were not explicitly included as predictors, their effects are implicitly captured in the regression coefficients. To address this issue, we state:

*"Changes in other meteorological forcings may also influence the estimation of eCH₄. These confounding drivers, such as solar radiation and wind speed, although they are considered to have minor impacts on the variations of eCH₄, were implicitly accounted for in the regression coefficients."*

How do the ensemble modeling results for the 2020 surge compare with other studies that used satellite data to interpret the surge? Authors mention Peng et al. 2022. In addition, Feng et al. 2023 (https://doi.org/10.5194/acp-23-4863-2023) and Qu et al 2022 (https://doi.org/10.1088/1748-9326/ac8754) attribute the surge to emissions, largely from wetland and water sources in Africa. A note on how your results agree or not would be useful given the attention in this manuscript and in the literature on the 2020 surge.

Response: Thank you for bringing up this important point. We have added statements in the manuscript to discuss the consistencies and discrepancies between our findings and those from the studies mentioned by the reviewer.

Our model ensemble suggests that a large portion of the methane increase in 2020 originates from the tropics, which aligns with the conclusions of Peng et al. (2022), Feng et al. (2023), and Qu et al. (2022). However, our results do not indicate the same magnitude of increase as reported by Peng et al. (2022). This difference may be partly attributed to uncertainty in the climate forcing inputs used in the model simulations—specifically precipitation data—where Peng et al. (2022) utilized three sets of reanalysis data, whereas our study use CRU and GSWP3-W5E5.

Additionally, there are discrepancies in the increase in wetland CH4 emissions between our study and those of Feng et al. (2023) and Qu et al. (2022). While our study suggest that Africa in 2020 has various change -0.2[-1.5-0.7] Tg CH4 yr-1. These discrepancies are primarily due to differences in methodology. Feng et al. (2023) and Qu et al. (2022) used atmospheric inversion with GOSAT satellite measurements as constraints. The GOSAT data over the tropics is limited by availability and is influenced by factors such as aerosols and clouds, which affect the accuracy of XCH4 estimates based on XCO2 measurements. In contrast, our process-based models cannot produce such high increase. This is an area that requires further investigation.

The strengthened discussion about 2020 surge is as follow: "The models consistently show that 2020 is the strongest positive anomaly year during 2000-2020, with a net increase of 2 [-2, 7] Tg $CH_4$ $yr^{-1}$ (mean [min, max]) in 2020 compared to 2019. This positive anomaly in 2020 (Table 1) is broadly consistent with a recent study (Peng et al., 2022) that reported 6±2.3 Tg $CH_4$ $yr^{-1}$ based on simulations of two bottom-up models with different climate datasets. The discrepancy in estimated magnitude between the Peng et al. (2022) and our results are partly due to the parameterizations of $CH_4$ module that causes lower annual magnitude in this study (~ 162±23 Tg $CH_4$ $yr^{-1}$ in 2020) compared to the Peng et al. (2022) study (177±31 Tg $CH_4$ $yr^{-1}$ in 2020). Additionally, the precipitation inputs in the climate forcing used in this study show a lower positive anomaly (~ of 20 mm $yr^{-1}$ in CRU over global wetland) in precipitation in 2020 compared to the reanalysis-based estimates (~ 40-117 mm $yr^{-1}$ over global wetland used in the study by Peng et al., (2022), which leads to lower estimates of wetland area and consequently lower emissions in this study. Moreover, our model ensemble does not indicate a strong increase (-0.2[-1.5-0.7] Tg $CH_4$ $yr^{-1}$) in $eCH_4$ in Africa in 2020. This contrasts with recent atmospheric inversions (Feng et al., 2023; Qu et al., 2023), which suggest a large increase of 11-17 Tg $CH_4$ $yr^{-1}$ above 2019 levels in African $CH_4$ emissions for 2020. The estimated increase from these inversions is equivalent to 55%-85% of total wetland CH4 emissions in Africa during 2010-2019 in our study (Figure 2). These discrepancies highlight the need for further studies to investigate the differences between these two approaches, including uncertainty in climate inputs in process-based bottom-up models and partitioning difference sources in atmospheric inversions.".

Minor comments

Line 52-53: This seems like a strong statement. I think this has been addressed, for example in inversions and in the authors' previous works, though perhaps not in the way it is addressed here. Consider being more specific.

Response: We have revised the sentence to *"However, despite reports of rising emission trends, a comprehensive evaluation and attribution of recent changes remains limited."*.

Line 55-56, "with an average decadal increase…": this sentence is a little unclear.

Response: We have revised the sentence to *"Our results estimated global average wetland $CH_4$ emissions at 158±24 (mean ± 1σ) Tg $CH_4$ $yr^{-1}$ over a total annual average wetland area of 8.0±2.0 $Mkm^2$ for the period 2010-2020, with an average increase of 6-7 Tg $CH_4$ $yr^{-1}$ in 2010-2019 compared to the average for 2000-2009."*

Line 97-98: Y Zhang et al. 2021 (https://doi.org/10.5194/acp-21-3643-2021), using GOSAT, is a useful comparison here.

Response: Thanks for providing the reference. We have cited the Zhang et al., 2021 in the text.

Line 136-137 "different prescribed parameters": Does this mean that each model has a different set of parameters and inputs, or that a different set of parameter values is given to each model? The current statement is vague.

Response: We have modified the statement to *"The prognostic wetland areal dynamics were independently determined by each model's hydrological modules, which use water table depth or soil moisture, combined with sub-grid topographic conditions to determine saturated areas within a land surface grid-cell (Zhang et al., 2016; Xi et al., 2021).".*

Line 144: Authors mention high correlations for the temperate region and high latitudes, but what about the tropics with the most emissions? Ensemble mean agreement with GIEMS2 in that region seems important, but it is not discussed and it is hard to tell the performance of the tropics from Figure S1.

Response: We have modified the statement to clarify as follow: *"The modeled temporal variations in wetland areas show high correlations with satellite-based products for temperate regions and high latitudes (Fig. S1), except in the tropics. The limited agreement in the tropics may be due to the influence of aerosols and clouds on satellite-based measurements, as well as the process-based model's performance limitations in representing wetland areas.".*

Line 205 and apparent Q10: Could authors comment on the choice ambient vs soil temperature here? Given the hysteresis effect, and evidence that methane emissions follow soil temperature rather than air temperature, soil seems the more logical choice, but I may be misunderstanding.

Response: Thank you for your insightful comment. All the models used in this study do indeed calculate soil temperature as part of their internal processes. However, for the purpose of unifying the analysis across models, we opted to use air temperature for a consistent comparison across models that may handle soil temperature dynamics differently.

Line 226, "Suggesting enhanced wetland-CH4 sensitivity under climate change": To me, authors haven't demonstrated that the larger IAV in the second decade considered is evidence of larger sensitivity under climate change. The statement may be true, but I

don't think authors have demonstrated it, so I suggest adjusting the statement or providing more evidence.

Response: We have revised the statement to *"The model ensemble demonstrates a consistent increase in interannual variability (IAV) in $\Delta eCH_4$ from 3.6±1.6 Tg CH$_4$ yr$^{-1}$ during 2000-2009 to 4.7±1.5 Tg CH$_4$ yr$^{-1}$ during 2010-2020, suggesting a potential increase in eCH$_4$ variability under climate change."*

Figure 2: Could authors add identifying markers for the regions in panels c,d to the maps?

Response: Thank you for the suggestion. We attempted to add markers to the maps a and b. However, after testing various options, we found that the markers cluttered the visual presentation and detracted from the clarity of the maps. To maintain readability, we decided not to include them. Instead, we have enhanced the borders between different regions to improve visualization.

Line 317-318, "with a range of -0.4 and 9.0 Tg…": Is this the distribution of coefficients among all the wetland models?

Response: Yes this is among all the wetland models. We have modified the sentence to clarify: *"The regression coefficients for $\gamma$ is 4.6 Tg CH$_4$ yr$^{-1}$ ℃$^{-1}$, with a range of -0.4 and 9.0 Tg CH$_4$ yr$^{-1}$ ℃$^{-1}$ between the 10$^{th}$ and 90$^{th}$ percentiles among all models."*.

Figure 3: It's unclear what the Gaussian density distribution curves represent, could more description be added to the caption? In panel b, the dashed lines are too faint to distinguish.

Response: The curves represent the probability distributions of the fitted parameters, derived from values of the individual models. The Gaussian distributions are fitted to show the range and central tendency of these sensitivity coefficients across the models. We have revised the figure caption for clarification: *"The curves represent the probability distributions of the sensitivity coefficients across the models, assuming a Gaussian distribution."* We've modified the dashed lines to be thinner for better virtualization.

Line 411-413, "Furthermore…eCh4": The meaning of this sentence is unclear.

Response: We have revised this sentence to clarify as follow: *"Furthermore, the modeled ensembles of prognostic wetland extents offer a complementary approach to satellite-based estimates (Prigent et al., 2020; Zhang, et al., 2021) and their impact on the spatial distribution of global eCH$_4$."*.

Lines 418-421: The MLR analysis seems to show a lower relative importance of the CO2 fertilization effect. Could authors reconcile the MLR analysis with the factorial analysis on this point?

Response: The MLR analysis of the $CO_2$ fertilization effect is consistent with the values calculated from the factorial analysis. The mean sensitivity coefficient $\beta$ is 0.18 Tg $CH_4$ $yr^{-1}$ $ppm^{-1}$, which corresponds to an approximate 2.3% increase relative to the annual total of 158 Tg $yr^{-1}$ under a 20 ppm increase in atmospheric $CO_2$ concentration. As suggested by the reviewers, we have revised the statement as follows: "The mean sensitivity coefficient $\beta$ and the results from the factorial experiment suggest a net increase in $eCH_4$ of 0.1%-2.3% relative to the annual total under an average ~20 ppm increase in atmospheric $CO_2$ concentration."

References:

Piao, S., Sitch, S., Ciais, P., Friedlingstein, P., Peylin, P., Wang, X., Ahlström, A., Anav, A., Canadell, J. G., Cong, N., Huntingford, C., Jung, M., Levis, S., Levy, P. E., Li, J., Lin, X., Lomas, M. R., Lu, M., Luo, Y., Ma, Y., Myneni, R. B., Poulter, B., Sun, Z., Wang, T., Viovy, N., Zaehle, S., and Zeng, N.: Evaluation of terrestrial carbon cycle models for their response to climate variability and to CO 2 trends, Global Change Biology, 19, 2117–2132, https://doi.org/10.1111/gcb.12187, 2013.

Zhao, F., Zeng, N., Asrar, G., Friedlingstein, P., Ito, A., Jain, A., Kalnay, E., Kato, E., Koven, C. D., Poulter, B., Rafique, R., Sitch, S., Shu, S., Stocker, B., Viovy, N., Wiltshire, A., and Zaehle, S.: Role of $CO_2$, climate and land use in regulating the seasonal amplitude increase of carbon fluxes in terrestrial ecosystems: a multimodel analysis, Biogeosciences, 13, 5121–5137, https://doi.org/10.5194/bg-13-5121-2016, 2016.

---

## Author Response (AR2)

RC1

Review Comments for egusphere-2024-1584
Title: Ensemble estimates of global wetland methane emissions over 2000-2020

Reviewer1: Thanks so much to authors for doing such a nice revision.

Only one technical point needs you to double check:

I see you added the mean CH4 values of the 2000-2010 and 2011-2020 in the Figure 1 caption. But the two horizontal lines you added, one across 6 on Y-axis and one across 0 on Y-axis, could bring a little confusion because they are not crossing at 152 and 158. And the 2000-2010 line is likely covered by the X-axis. Maybe you can consider adding a second Y-axis, or just mention the two mean values without adding the two lines? Or perhaps the caption itself already explains well, and no need to change anything.

Response: Thank you for the positive comments. We appreciate your recognition of the significance of our findings. In response to the reviewer's comments, we have modified Figure 1a to use absolute values instead of $\Delta eCH_4$ to avoid confusion. Please see the revised Figure 1 below.

[Figure]

Figure 1: Simulated global wetland $CH_4$ emissions from the model ensemble for 2000-2020. a, Time series of annual total emissions during 2000-2020, with the shaded area

representing the range between minimum and maximum modeled emissions. The horizontal lines represent the ensemble means of 2000-2009 (152 Tg $CH_4$ $yr^{-1}$) and 2010-2019 (158 Tg $CH_4$ $yr^{-1}$), respectively. b, Latitudinal gradient of $eCH_4$ difference ($\Delta eCH_4$), with the mean annual total $\Delta eCH_4$ for each of the 30° latitude bins from the two sets of simulations shown. The change is calculated relative to the mean of the 2000-2009 level from the two sets of simulations with prognostic wetland emission models grouped by different climate datasets, CRU and GSWP3-W5E5. c, Boxplots of mean seasonal $\Delta eCH_4$ for the three regions. The central mark and the bottom and top edges of the box indicate the median, and the 25th and 75th percentiles of the ensemble, respectively. The colored lines represent the average seasonal cycle of 2000-2009 from the simulations grouped by two climate datasets, CRU and GSWP3-W5E5.